# Metabolic Pathway Pairwise-Based Signature as a Potential Non-Invasive Diagnostic Marker in Alzheimer’s Disease Patients

**DOI:** 10.3390/genes14061285

**Published:** 2023-06-17

**Authors:** Yunwen Feng, Xingyu Chen, Xiaohua Douglas Zhang, Chen Huang

**Affiliations:** 1Dr. Neher’s Biophysics Laboratory for Innovative Drug Discovery, State Key Laboratory of Quality Research in Chinese Medicine, Macau University of Science and Technology, Taipa, Macao SAR 999078, China; morphy1678@outlook.com (Y.F.); xiaozeshiwoxiaodi@outlook.com (X.C.); 2Department of Biostatitics, College of Public Health, University of Kentucky, Lexington, KY 40536, USA

**Keywords:** Alzheimer’s disease, noninvasive diagnosis, metabolic abnormalities, biomarker, peripheral blood, multi-machine learning

## Abstract

Alzheimer’s disease (AD) is an incurable neurodegenerative disorder. Early screening, particularly in blood plasma, has been demonstrated as a promising approach to the diagnosis and prevention of AD. In addition, metabolic dysfunction has been demonstrated to be closely related to AD, which might be reflected in the whole blood transcriptome. Hence, we hypothesized that the establishment of a diagnostic model based on the metabolic signatures of blood is a workable strategy. To that end, we initially constructed metabolic pathway pairwise (MPP) signatures to characterize the interplay among metabolic pathways. Then, a series of bioinformatic methodologies, e.g., differential expression analysis, functional enrichment analysis, network analysis, etc., were used to investigate the molecular mechanism behind AD. Moreover, an unsupervised clustering analysis based on the MPP signature profile via the Non-Negative Matrix Factorization (NMF) algorithm was utilized to stratify AD patients. Finally, aimed at distinguishing AD patients from non-AD groups, a metabolic pathway-pairwise scoring system (MPPSS) was established using multi-machine learning methods. As a result, many metabolic pathways correlated to AD were disclosed, including oxidative phosphorylation, fatty acid biosynthesis, etc. NMF clustering analysis divided AD patients into two subgroups (S1 and S2), which exhibit distinct activities of metabolism and immunity. Typically, oxidative phosphorylation in S2 exhibits a lower activity than that in S1 and non-AD group, suggesting the patients in S2 might possess a more compromised brain metabolism. Additionally, immune infiltration analysis showed that the patients in S2 might have phenomena of immune suppression compared with S1 and the non-AD group. These findings indicated that S2 probably has a more severe progression of AD. Finally, MPPSS could achieve an AUC of 0.73 (95%CI: 0.70, 0.77) in the training dataset, 0.71 (95%CI: 0.65, 0.77) in the testing dataset, and an AUC of 0.99 (95%CI: 0.96, 1.00) in one external validation dataset. Overall, our study successfully established a novel metabolism-based scoring system for AD diagnosis using the blood transcriptome and provided new insight into the molecular mechanism of metabolic dysfunction implicated in AD.

## 1. Introduction

Alzheimer’s disease (AD) is an extremely common neurodegenerative disease that is the leading cause of dementia. It typically begins with a deterioration in memory and is characterized by a progressive decline in cognitive function [1]. With the aging of the population and longer lifespans, the incidence of the disease continues to rise. There are approximately 50 million people worldwide with AD [2], and this number is expected to increase rapidly in the coming decades. Currently, there is no curative treatment for AD, and the best therapy is early diagnosis and the delay of disease progression [3]. Therefore, AD risk prediction is in urgent need of effective biomarkers.

The diagnosis of AD involves a variety of methods, including clinical presentation, cognitive tests, brain imaging, cerebrospinal fluid analysis, and blood testing. Clinical presentation involves observing the patient’s symptoms, including cognitive and memory impairment as well as behavioral and emotional changes. Cognitive tests, such as the Mini-Mental State Examination (MMSE) and Montreal Cognitive Assessment (MoCA), are used to evaluate a patient’s cognitive ability. Brain imaging techniques, such as Positron Emission Tomography (PET) and Magnetic Resonance Imaging (MRI) scans, can reveal structural and functional changes in the brain, i.e., brain atrophy and the accumulation of β-amyloid plaques. However, these diagnostic methods are time-consuming, costly, and subjective based on the clinic doctors’ experience [4]. Particularly, the US National Institute on Aging and the Alzheimer’s Association proposed using biomarkers as a purely biological definition of AD [5]. For example, cerebrospinal fluid examination (CSF) can detect the accumulation of β-amyloid protein plaques and other biomarkers associated with AD, i.e., Aβ42, T-tau, and P-tau [6]. Although the CSF test is effective for AD, its highly invasive character remains challenging for AD patients, especially elderly patients. More importantly, the establishment of reliable biomarkers based on CSF core biomarkers, i.e., Aβ and tau, has culminated in a debate derived from conflicting results and theories [7]. It is urgent, thereby, to identify novel biomarkers for early diagnosis of AD as well as potential targets for therapeutic methods in AD. Recently, accumulating evidence indicated that the detection of fluid biomarkers from blood as diagnostic tools for AD is definitely a practical solution [8,9]. Blood testing detects specific proteins or other biomarkers in the blood and thus can be used to predict a patient’s risk of developing AD. This approach has the advantage of being convenient, fast, and non-invasive compared with other methods for AD diagnosis.

Although the exact cause of AD is still not fully understood, many studies have suggested that metabolic abnormalities are associated with the development of AD [10]. There has been growing interest in the role of metabolic dysfunction, particularly lipid, glucose, and energy metabolism, in the development and progression of AD [11,12,13]. Abnormalities in lipid metabolism in AD refer to a series of aspects, including high cholesterol, high triglycerides, and low-density lipoprotein [14]. These abnormalities can lead to atherosclerosis and cardiovascular disease [15], greatly increasing the risk of patients developing AD. Some studies have further demonstrated that high cholesterol may lead to the formation of β-amyloid protein plaques, which are one of the typical features of AD. β-amyloid protein plaques can damage neurons in the brain, leading to cognitive impairment, memory loss, and neuronal death [16]. Apolipoprotein E (*ApoE*) is an important lipid metabolism gene. When it performs its normal function, it can repair synapses and maintain neuronal structure [17]. However, when its structure is disrupted, it can affect these functions. Therefore, the *ApoE* gene has been identified as a major risk factor for AD [18]. Currently, many studies also indicate that glucose and energy metabolism are significantly associated with AD, such as the tricarboxylic acid (TCA) cycle, oxidative phosphorylation deficits, and pentose phosphate pathway impairment [19]. Glucose is an important energy substrate for brain and neurons in brain need a great quantity of energy to sustain the normal activity [19]. However, a decrease in glucose and energy metabolism is also observed in AD patients by research [20]. In addition, oxygen and glucose metabolic rates are significantly changed in AD because of the alterations in the glycolytic pathway and TCA cycle [21]. Reasonably, abnormalities of metabolism exhibit a close association with the onset and progression of AD, and the identification of novel metabolism-related biomarkers is a workable strategy for the diagnosis of AD.

In the present study, we hypothesized that molecular metabolism abnormalities in AD might reflect in the metabolic gene expression of peripheral blood, and characterizing those unconventionally metabolic genes in blood may give rise to a promising non-invasive biomarker for the diagnosis of AD, particularly in the early stages. Initially, we unveiled the difference in peripheral blood gene expression between AD and non-AD patients based on the high-throughput RNA sequencing data, along with the relevant biological processes and pathways involved. Subsequently, inspired by Lixin Cheng et al.’s study [22], we proposed a novel approach to quantifying the difference between a pair of metabolic pathways within each individual sample (including AD and non-AD patients). The main merit of this approach is that it can well avoid the batch effect derived from different datasets. This analysis successfully figured out several metabolic pathway pairwise (MPP) signatures associated with AD. Furthermore, all the AD patients could be classified into two subgroups via the unsupervised clustering analysis based on the MPP signature matrix, which exhibits distinct patterns of immunity and metabolism. Eventually, we utilized multiple machine learning methods to screen out key MPP signatures correlated to AD and establish a metabolic pathway pairwise scoring system (MPPSS) for AD diagnosis (Figure 1). The MPPSS achieved a high AUC in not only the test data but also the independent validation datasets. In conclusion, using the whole blood transcriptome, we have developed a reliable and sensitive scoring system for non-invasive diagnosis and intervention of AD. This system holds significant potential value in clinical usage as well as in studying the interrelationships between metabolic pairwise pathways and the high-risk AD population.

## 2. Materials and Methods

### 2.1. Data Acquisition and Preprocess

Eight gene expression datasets of AD patients and non-AD controls are obtained from the Gene Expression Omnibus (GEO) database (https://www.ncbi.nlm.nih.gov/geo/query/acc.cgi, accessed on 26 April 2023) [17]. The basic statistics of relevant information, including platform, tissue, and sample number, were summarized in Appendix A. Of them, three datasets were merged and used for downstream analysis as well as the establishment of a model for AD diagnosis. Merged data, referred to here as metaGEO, consisted of GSE140829 (249 Control and 204 AD patients), GSE63060 (104 Control and 145 AD patients), and GSE63061 (234 Control and 238 AD patients). The R package “SVA” [18] was applied to remove batch effects among different datasets by the ComBat() function (Appendix A). Patient ID, gender, race, age, and APoE stage among the clinical data of AD and non-AD samples and the association between each clinic variable and AD disease status based on single variable analysis were summarized in Table 1. The remaining datasets (GSE97760, GSE148822, GSE163877, and GSE104704) were used as independent data for the validation of the diagnosis model.

### 2.2. Construction of MPP Signatures

The Kyoto Encyclopedia of Genes and Genomes (KEGG) database (https://www.genome.jp/kegg/pathway.html, accessed on 26 April 2023) collects different types of biological pathways including metabolism, genetic information processing, environmental information processing, etc. [23]. In our study, we selected all 84 pathways in the category of metabolism, including subcategories such as carbohydrate metabolism, energy metabolism, lipid metabolism, etc. (Appendix A). After removing the batch effect from 975 samples, the single sample gene set enrichment analysis (ssGSEA) score was used to sort transcriptome data and assign different weights based on their positions in the gene set in order to calculate the enrichment level of specific pathways in the sample. The ssGSEA method was derived from the R package “GSVA” [24], and herein it was utilized to evaluate the pathway activity of 84 metabolic pathways (Appendix A). After that, we perform subtraction between two pathways and iterate through all 84 KEGG pathways, resulting in 3486 MPP signatures at the end. By subtracting the Metabolism Pathway (MP) values in twos, we can suggest possible interaction relationships between pathways. We constructed the MPP signatures based on Lixin Cheng et al.’s study [22] and displayed this process below.

#### 2.2.1. Within-Sample Analysis

We performed the comparisons for MPP signatures based on the ssGSEA value of the single pathway. For each sample, the MP score of the k-th sample was recorded as the vector.
$$S^{\left(k\right)}=\left({\mathrm{MP}}_{1}^{\left(k\right)},{\mathrm{MP}}_{2}^{\left(k\right)},\;\ldots ,{\;\mathrm{MP}}_{m}^{\left(k\right)}\right)$$
where MP_m_ represents the ssGSEA score of the m-th metabolic pathway and the superscript k indicates the k-th sample. Then, we defined the relative value of a MP pair, which can be summarized in the MPP signature as
$$r_{\mathrm{ij}}^{\left(k\right)}=I({\mathrm{MP}}_{i}^{\left(k\right)}-{\mathrm{MP}}_{j}^{\left(k\right)})$$
where $I\left(x\right)=\{\begin{array}{cc} 1, & \mathrm{if}\;x\geq 0\\ -1, & \mathrm{if}\;x<0 \end{array}$ is an indicating function that denotes whether x is larger than or less than 0. If MP is larger or equal to MP, the relative value of the MPP signature is assigned as 1. If not, the relative value is −1. For each MPPS $\left({\mathrm{MP}}_{i}^{\left(k\right)},{\mathrm{MP}}_{j}^{\left(k\right)}\right)$, subtraction was used to transform the discrete value, which was represented as ${\mathrm{MP}}_{i}^{\left(k\right)}-{\;\mathrm{MP}}_{j}^{\left(k\right)}$, $\forall \;i,j\;\in \left(1,\ldots ,m\right),\;i\neq j$. $R_{\mathrm{MP}}^{\left(k\right)}$ was a vector comprised of the pathway activity values of all pairs of MPs within the k-th sample.
$$R_{\mathrm{MP}}^{\left(k\right)}=\left(r_{12}^{\left(k\right)},r_{13}^{\left(k\right)},r_{14}^{\left(k\right)},\ldots ,r_{1m}^{\left(k\right)},r_{23}^{\left(k\right)},r_{24}^{\left(k\right)},\ldots ,r_{2m}^{\left(k\right)},\ldots r_{\left(m-1\right)m}^{\left(k\right)}\right)$$

Since there were $C_{m}{}^{2}=\frac{m!}{2!\left(m-2\right)!}$ MPP signatures, $R_{\mathrm{MP}}{}^{\left(k\right)}$ were in $C_{m}$ dimensions.

#### 2.2.2. Cross-Sample Analysis

Within-sample calculation was followed by cross-sample analysis between AD and non-AD groups. To acquire those significantly differential MPP signatures, we conducted the count test and thus compared the number of $r_{\mathrm{ij}}{}^{\left(k\right)}=1\left({\mathrm{MP}}_{i}{}^{\left(k\right)}\geq {\mathrm{MP}}_{j}{}^{\left(k\right)}\right)$ and $r_{\mathrm{ij}}{}^{\left(k\right)}=-1\left({\mathrm{MP}}_{i}{}^{\left(k\right)}\geq {\mathrm{MP}}_{j}{}^{\left(k\right)}\right)$ in AD and non-AD groups. Finally, the contingency table (Table 2) of MPP signatures for AD and non-AD samples was shown as follows:

Subsequently, the chi-square test was calculated based on this contingency table to quantify the association between AD and each MPP signature. Holm’s adjustment was used for multiple comparisons. The relevant analysis above was conducted using R version 4.2.1 (https://www.r-project.org/). After gaining these MPP signatures, we used Cytoscape software V3.9.1 [25] (http://www.cytoscape.org/) to analyze and visualize the metabolic network and thus detect hub nodes by the Maximal Clique Centrality (MCC) method in cytoHubba plug-in application (V0.1) of the Cytoscape [26].

### 2.3. Unsupervised Clustering to Characterize AD Patient Subgroups

A total of 112 significantly AD-related MPP signatures were detected via the above-mentioned cross-sample analysis between AD and non-AD groups based on the MPP signature matrix (3486 MPP signature × 487 AD patients) of the metaGEO cohort. Then those significantly AD-related MPP signatures were subjected to unsupervised clustering analysis using the non-negative matrix factorization (NMF) algorithm. The NMF [27] algorithm has been widely applied to dimensional reduction and clustering on non-negative data. The distinct difference between NMF and other unsupervised clustering methods such as principal component analysis (PCA) is the non-negative constraints. In addition, the NMF method exhibits better clustering performance in gene expression data compared with other unsupervised clustering methods [28]. Therefore, NMF is an efficient method for identifying distinct molecular subtypes via gene expression data. Based on the integration of MPP signatures, we used the “NMF” R package [29] to perform NMF clustering. Before clustering, we take the exponent of constant e to eliminate negative numbers in the matrix. In the present study for the NMF method, the standard “brunet” method was selected and the “nrun” parameter set to 10 for NMF clustering. Multiple runs are essential to yielding a stable and robust clustering outcome in AD patients. After iterations of the consensus clustering algorithm, the number of optimal clusters was confirmed according to the rockfall diagram (Appendix A). The cophenetic, dispersion, and silhouette indicators determined that the optimal clustering number is 2.

### 2.4. Establishment of AD Diagnostic MPPSS by Using Multiple Machine Learning Approaches

Next, we established machine learning-based MPPSS for the diagnosis of AD and non-AD samples. We used five artificial intelligence frameworks and conducted AD diagnosis model establishment and validation. Finally, we constructed AD diagnostic MPPSS by using the best performing method among the five machine learning methods, including extreme gradient boosting (XGBoost [30], R Package “xgboost”, V4.2.2, https://xgboost.ai/, accessed on 24 April 2023), Boruta (R Package “Boruta” [31], V8.0.0, https://gitlab.com/mbq/Boruta/, accessed on 12 April 2023), random forest (R Package “randomForest”, V4.7-1.1), decision tree (R Package, “rpart” [32], V4.1.16, https://cran.r-project.org/web/packages/rpart/index.html, accessed on 24 April 2023), LASSO (R Package “glmnet” [33] V4.1-4, https://cran.r-project.org/web/packages/glmnet/index.html, accessed on 12 April 2023). Boruta is a feature selection algorithm based on random forests that can identify significant and insignificant features, helping to determine which features are useful for the model [31]. XGBoost is an enhanced learning algorithm based on decision trees that can also be used for feature selection [30]. LASSO is a regression analysis method that can reduce the number of parameters in the model by adding penalty terms and selecting key features [34]. The principle of lasso regression is to add an L1 regularization term that limits the number of features in the model, thereby reducing the risk of overfitting. Specifically, the L1 regularization term adds the sum of the absolute values of the regression coefficients to the loss function, which forces some of the feature coefficients to be zero during optimization and achieves a feature selection effect. Decision tree and random forest are also common feature selection algorithms that can rank the importance of each feature through calculation [35]. We comprehensively evaluated the performance of five machine learning methods based on 10-fold cross-validation via multiple indexes (AUC, recall, precision, etc.). The LASSO method outperformed other methods. In summary, the metaGEO dataset was randomly split into a training set (comprising 70% of the total dataset) and a testing set (comprising 30% of the total dataset) for the purpose of model establishment and accuracy validation [36]. Subsequently, we established an ordinary binary (AD and non-AD) logistic regression with LASSO. The “cv.glmnet” function was used to select the optimal lamda parameter. To construct a reliable AD diagnosis model, we repeated the selection features and model prediction with one thousand iterations and decreased the influence caused by the occasional confusing factor. To validate the LASSO model in training and testing datasets, the AD risk score of each sample was calculated according to the following formula. The coef represents the risk coefficient of each MPP signature, and MPPS_i_ is the activity of the i-th MPP signature.
$$\mathrm{risk}\;\mathrm{score}=\sum_{i=1}^{n}\mathrm{coef}\left({\mathrm{MPPS}}_{i}\right){*\mathrm{MPPS}}_{i}$$

For the four independent validation datasets, the established model based on the metaGEO dataset was used to calculate the risk score for each sample, and test the LASSO model’s performance via the AUC index. All the source code for reproducing the examples presented in this paper can be consulted in the following GitHub repository: https://github.com/ChenHuangMUST/MPP-signature-for-AD-diagnosis.git (accessed on 7 May 2023).

### 2.5. Immune Infiltration Analysis by the CIBERSORT Algorithm

As a machine learning method based on the linear support vector regression method of calculating cell fractions from gene expression data, Cell-type Identification By Estimating Relative Subsets Of RNA Transcripts (CIBERSORT) [37], could infer the proportions of B cells, plasma cells, T cells, natural killer cells, monocytes, macrophages, dendritic cells, mast cells, eosinophils, and neutrophils that had infiltrated among 3 groups (S1, S2, and non-AD groups). By using the CIBERSORT algorithm, we analyzed the AD patients’ gene expression data and quantified the relative proportions of 22 infiltrating immune cells.

### 2.6. Gene Differential Expression Analysis and Functional Annotation Analysis between the AD and Non-AD Groups, as well as within the Two AD Subgroups

We identified the differentially expressed genes (DEGs) by comparing non-AD and AD groups, as well as AD subgroups, using the “limma” R package [38]. It developed a multidimensional gene expression analysis method to evaluate the differences in gene expression and identify which biological pathways are affected by these differences. First, we converted the transcriptome data downloaded from GEO from probe-level format to gene-level format for further analysis. Subsequently, the data underwent batch effect removal, and differential expression analysis was performed between the AD and non-AD samples, as well as within the two AD subgroups. The logarithmic fold change (logFC) values that exceeded the limit of the mean value +/− two folds of standard deviation and the adjusted *p* value less than 0.05 were considered DEGs. A volcano plot of the DEGs was created using the “ggplot2” R package. After obtaining the DEGs, we performed functional enrichment analysis based on the Gene Ontology (GO) knowledgebase [39], the KEGG database, and the Hallmarks pathway derived from the Molecular Signatures Database (MSigDB) [40] via the “GSEA” and “clusterProfiler” R packages. In addition to GO and KEGG pathways, Hallmark pathways are a classification system used to describe important biological processes related to disease development.

## 3. Results

### 3.1. Comparative Transcriptome Analysis Characterizes Metabolic Hallmarks of Peripheral Blood in AD

After integration of three GEO datasets, referred to as metaGEO, the differential gene expression analysis based on blood RNA-seq data between 488 AD and 487 non-AD samples was performed, which yielded 605 DEGs (324 up-regulated and 281 down-regulated DEGs) (Figure 2a,g, and Appendix A). Upregulated genes such as *KLRB1*, involved in immune regulation, and *HINT7*, involved in neurodevelopment, play critical roles in these processes. Protein-coding genes, such as *LSM3*, *ATPO5*, *COX7*, and *RPL17*, participate in mitochondrial respiration and RNA splicing. In contrast, downregulated genes such as *NBEAL2*, *CEBPD*, *OSCAR*, *CD14*, *PAD14*, *CRISPLD2*, and *PGLYRP1* are associated with immune response and inflammation. *HK3*, *STXBP2*, and *PGD* are related to metabolic processes. Genes such as *APBB1IP*, *ITPK1*, and *TLN1* are involved in signal transduction and neurodevelopment. Subsequently, we conducted cluster analysis of GO, KEGG, and hallmark pathways. Using GSEA to sort the logFC of DEGs, we found that in the AD and non-AD groups, DEGs were mainly enriched in KEGG pathways (Figure 2c and Appendix A) related to nucleotide excision repair, citrate cycle (TCA cycle), pyruvate metabolism, drug metabolism-cytochrome P450, oxidative phosphorylation, and ECM-receptor interaction. GO was enriched in chemical stimulus perception, aerobic respiration, oxidative phosphorylation, electron transport-coupled ATP synthesis, sensory perception, etc. (Figure 2d and Appendix A). Hallmark pathways were enriched in biological processes such as oxidative phosphorylation, MYC V1 targets, and heme metabolism (Figure 2f and Appendix A). We further depicted the differences in metabolic pathways, including lipid, glucose, and energy metabolism, between AD and non-AD samples (Figure 2e). We performed a differential MPP signature analysis between AD and non-AD groups via the chi-square test (adjusted *p* value < 0.01) and obtained 112 significantly differential MPP signatures (Figure 2h and Appendix A). Then, we processed network analysis for those MPP signatures to detect key metabolic pathways (Appendix A). Among them, several glucose and lipid metabolic pathways, including galactose metabolism, biosynthesis of unsaturated fatty acids, and arachidonic acid metabolism, are identified (Figure 2b and Appendix A). Oxidative phosphorylation was consistent with our previous differential gene expression analysis between AD and non-AD groups (Figure 2c,f).

### 3.2. NMF Clustering Analysis of AD Patients Based on Peripheral Blood MPP Signatures Reveals Distinct Patterns of Lipid, Glucose, and Energy Metabolism

To quantify the degree of metabolic differences in AD patients, we selected 112 MPP signatures via chi-square test (adjusted *p* value < 0.01) for NMF clustering (Figure 3a). NMF clustering of the MPP signature matrix revealed two main AD clusters, referred to here as S1 and S2 (with 295 and 193 cases, respectively; Table 1). Then, we conducted the differential gene expression analysis of blood-based RNA-seq data between the S1 and S2 groups, which yielded 675 DEGs (420 up-regulated and 255 down-regulated DEGs; Figure 3b,d, and Appendix A). As a result, upregulated genes such as *ANAX1* participate in biological functions such as autophagy and metabolic regulation. *ARGLU1* is mainly involved in metabolic regulation. *S100A8* and *KLRB1* are involved in inflammation and immune regulation. *TOMM7* dominates cerebrovascular network homeostasis, and some studies have found that it is associated with the neurodegenerative disease Parkinson’s [41]. Downregulated genes such as *LSP1* and *CDA* are involved in immunity. *SLC25A37* participates in metabolic development. *NINJ2* and *GRINA* regulate neurodevelopment, while *EPB49* is involved in cell extension and development. DEGs were enriched in KEGG pathways including neuroactive ligand-receptor interaction, cytokine-cytokine receptor interaction, olfactory transduction, etc. (Figure 3e and Appendix A). Between the S1 and S2 groups, GO enrichment of DEGs was found to be involved in autophagy, sensory perception, macroautophagy, organelle catabolic processes, chemical stimulus, olfaction, and stimulus detection in sensory perception, etc. (Figure 3f and Appendix A). DEGs were mainly enriched in hallmark pathways consisting of glycolysis, apoptosis, fatty acid metabolism, estrogen response, and oxidative phosphorylation (Figure 3c and Appendix A). Similarly, the chi-square test was used to detect differential MPP signatures (adjusted *p* value < 0.01) between the S1 and S2 groups, which generated 120 differential MPP signatures (Figure 3h and Appendix A). Subsequent network analysis for those differential MPP signatures was conducted to screen out key metabolic pathways (Appendix A). As a result, many key metabolic pathways correlated to AD were detected (Figure 3g and Appendix A), including α-Linolenic acid metabolism, fructose and mannose metabolism, fatty acid elongation, etc. Additionally, several pathways such as oxidative phosphorylation, biosynthesis of unsaturated fatty acids, and drug metabolism-cytochrome P450 were enriched in our previous functional analysis of DEGs between the S1 and S2 groups (Figure 3c and Appendix A), implying the reliability of these AD-related metabolic pathways. We compared metabolic pathway activity between the S1 and S2 groups via ssGSEA analysis. The activity of oxidative phosphorylation in the S1 group is higher than that in the S2 group, whereas the activities of most lipid metabolisms, i.e., α-Linolenic acid metabolism, arachidonic acid metabolism, and ether lipid metabolism, in the S1 group were found to be lower than those in the S2 group (Figure 3j).

### 3.3. Comprehensive Evaluation of Immune Cell Infiltration Characteristics in AD Subgroups and the Non-AD Control Group

Immune cell infiltration is significantly associated with AD progress, and it is one of the hallmarks of AD [42]. To comprehensively evaluate the immune infiltration characteristics of two AD subgroups and one control group, we applied the CIBERSORT algorithm to estimate the proportions of 12 immune cell types (i.e., B cells, plasma cells, CD8+ T cells, CD4+ T cells, γδ T cells, NK cells, monocytes, macrophages, dendritic cells, mast cells, eosinophils, and neutrophils) in each sample. The results presented significantly different immune infiltration features in the subgroups, indicating that the patients in these groups had different immune cell infiltration landscapes. As shown in Figure 3i, T cell memory activation and T cell CD4 memory resting were significantly lower in S2 patients, while T cell CD4 naive infiltration was significantly higher in S1 patients. Dendritic cells activated and dendric cells resting were both extremely significant in S2. The formal is higher than the two other groups, and the latter is on the opposite side. While T cells were regulatory, the index of S2 group immune infiltration was obviously higher than the other groups, whereas the control group was the second. In T cells γδ immune infiltration, the S2 group had the lowest proportion compared with the S1 group and the non-AD group. Mast cell activation was significantly accumulated in S2, while the mast cell resting in S2 is on the counterpart.

### 3.4. Establishment of MPPSS for Distinguishing AD Patients from Non-AD Patients

Initially, the metaGEO dataset was used to construct the AD diagnostic model. To select the optimal machine learning method, five machine learning methods were separately applied to establish an MPSS classifier on the metaGEO data, including Boruta, XGBoost, random forest, decision tree, and LASSO (Figure 4a). Additionally, we used 10-fold cross-validation to evaluate the performance of models on AD diagnosis. The MPP signature matrix was subjected to each machine learning model for 10-fold cross-validation with 100 iterations, respectively (Appendix A). We evaluated the performance of the MPSS classifier using the ROC metric, etc., with the points on the ROC curve representing its performance at different classification thresholds. The results indicated that the LASSO model, referred to as the MPP scoring system (MPPSS), outperformed other models on the metaGEO dataset (Figure 4b), which consists of 13 important MPP signatures related to AD (Figure 4c and Table 3). These pathways are closely related to biological processes such as oxidative phosphorylation, pyruvate metabolism, or heme metabolism and involve the synthesis, degradation, and transformation of various important metabolites, such as steroid hormones, GPI anchors, heparan sulfate/dermatan sulfate, terpenoid backbones, etc. Understanding these pathways can help us better understand the structure and function of metabolic regulatory networks, providing insights into disease prevention and treatment.

Next, we found that the AUC value of MPPSS for the training set was 0.73 (95%CI: 0.70, 0.77), and for the testing set, it was 0.71 (95%CI: 0.65, 0.77; Figure 4d). The performance of the training and validation sets directly demonstrates the predictive potential of the MPPSS. Then, to further evaluate the robustness and accuracy of the MPPSS, we validated it using four independent GEO datasets. Our MPPSS had good performance for predictions on the independent GEO sets. Concretely, when evaluating the model on the whole blood validation sets (GSE97760), we obtained an AUC value of 0.99 (95%CI: 0.96, 1.00; Figure 4e). The AUC values are likewise higher than 0.7 in the resting brain tissue validation sets (GSE148822 and GSE104704; Figure 4e). In addition, to evaluate the classification performance of the MPPSS, we compared the APoE genotypes and metabolic pathways in two groups divided by the MPPSS on the metaGEO dataset (Figure 4f). and the results indicated that lipid (α-Linolenic acid metabolism and biosynthesis of unsaturated fatty acids), glucose (galactose metabolism and pentose phosphate pathway), and energy (oxidative phosphorylation) metabolism are significantly different (Figure 4g).

## 4. Discussion

AD is an incurable neurodegenerative disorder associated with aging, and its underlying mechanisms are not yet fully understood [43]. Early diagnosis and delay of the disease process are regarded as the best treatments for AD. In our study we aimed to develop a diagnostic scoring system (MPPSS) for AD patients based on blood gene expression data. The advantages of blood-based biomarker diagnosis for AD include its non-invasiveness, safety, ease of use, low cost, and high accuracy compared to other traditional diagnostic methods, which require invasive procedures such as lumbar puncture or intracranial injection to collect samples.

Recent studies suggest that metabolic pathways including lipid, glucose, and energy metabolism may play a role in the development of AD [19,44]. Therefore, we established MPP signatures for the characterization of the interplay between metabolic pathway pairs. Based on the MPP signatures, we identified two subsets (S1 and S2) of AD patients via NMF clustering. In the S1, S2, and non-AD groups, the down-regulated genes are mostly related to immunity, neurogenesis, and signal transduction, while the up-regulated genes are mostly related to mitochondrial respiration and RNA splicing. Furthermore, we conducted the immune infiltration analysis for three groups and found that the S2 group had a lower immune proportion, which might suggest a strong correlation between AD progression and immunity. Finally, we constructed MPPSS for the AD diagnosis. Compared with a single marker-based diagnostic model, the MPP signature-based diagnostic model has more power for characterization of the interaction among metabolic pathways in AD onset and development. The MPPSS holds considerable potential for assisting doctors in diagnosing elderly patients. It also suggests that MPP signatures may be used as diagnostic biomarkers in clinics.

Overall, these findings suggest that metabolic pathways may provide potential diagnostic biomarkers for AD, particularly through blood-based analysis. Moreover, the involvement of cytochromes P450 in lipid homeostasis and detoxification processes further supports the role of metabolism in AD development [45]. Many studies have shown cytochromes P450 of the liver are involved in the maintenance of lipid homeostasis, such as cholesterol, vitamin D, oxysterol, and bile acid metabolism [46,47]. In detoxification processes of endogenous compounds such as bile acids [48]. The correlation provides evidence in support of our research findings. The core metabolic network metabolism of xenobiotics by cytochrome P450 (hsa00980) and drug metabolism by cytochrome P450 (hsa00982) are involved in the metabolic mechanisms associated with cytochrome P450. The metabolism of xenobiotics by cytochrome P450 appeared as an important core metabolic pathway in both the comparison of AD vs. non-AD (Figure 2b) and S1 vs. S2 (Figure 3g), and drug metabolism-cytochrome P450 appeared in S1 vs. S2 (Figure 3g) individually.

The activity of cytochrome P450 protein is also regulated by the lipid environment [49]. The lipid level may have an important impact on the onset and development of AD [50,51]. In our study, the differential enrichment of lipid metabolism pathways such as steroid biosynthesis, sphingolipid metabolism, glycerolipid metabolism, etc. (Figure 2e) supported this point of view.

Alzheimer’s disease is believed to be caused by reactive oxidative stress (ROS), which occurs prior to the formation of Aβ-plaques and neurofibrillary tangles [52]. The core metabolism pathway, that is, the biosynthesis of unsaturated fatty acids (hsa01040), identified in the present study has been demonstrated to be associated with ROS production [53]. Another core metabolism involved in the metabolism of unsaturated fatty acids was reported to be considerably disrupted in the brains of individuals with different levels of Alzheimer’s pathology [54]. What is more, cysteine and methionine metabolism (hsa00270) also plays an essential role in ROS; it can be oxidized and has been implicated in caloric restriction and aging [55]. These results are shown in Figure 2b.

Among these metabolism pathways, oxidative phosphorylation (Figure 2e) plays a crucial role in brain cell energy metabolism [56] and has been shown to be involved in the pathogenesis of AD [57]. Other pathways, including pyruvate metabolism [58], porphyrin metabolism [59] (Figure 2e), and fatty acid biosynthesis [46] (Figure 3e), have also been found to be implicated in AD. The dysregulation of these pathways may lead to cellular energy metabolism disruption, oxidative stress, and cell death, which may negatively affect the occurrence and development of AD [46,58,59,60].

Through analyzing the proportions of different immune cells in whole blood, a better understanding of the pathogenesis of AD can be gained. For example, inflammation may be an important trigger for AD, and certain immune cells such as macrophages and T cells are associated with inflammation. The comparison of T cells and macrophages among the three groups demonstrated that the AD patients in S2 had low accumulation. T cell memory activation and T cell CD4 memory resting were significantly lower in S2 patients, while T cell CD4 naïve infiltration was significantly higher than that of S1 patients. Memory T cells are a subset of T cells that can encounter foreign substances and antigens and become activated more effectively; in the meantime, CD4 T cells help coordinate immune responses by releasing cytokines and other signaling molecules [61], implying the patients in S2 exhibit lower immunity. There was a significant difference in the level of γδ immune infiltration among the S2 group compared to the other groups, with the S2 group exhibiting the lowest level. T cells with γδ receptors form a small percentage of lymphocytes in healthy individuals, whereas their number increases in people with immunological disorders. Additionally, we found the patients in S2 possessed the highest proportion of regulatory T cells (Treg), which is a hallmark of immunological suppression These findings suggested that the patients in S2 might have a more severe progression of AD, as well as the high confidence and clinical significance of AD subgrouping derived from NMF clustering.

Additionally, there are significant differences in the enrichment of mast cells among the three groups. Concretely, mast cell activation was significantly higher in the S2 group, while mast cell resting in the S2 group is on the counterpart. Derived from the myeloid lineage, mast cells are a category of immune cells that exist in connective tissues across the body [62]. Fibrillar Aβ peptides are known to play a significant role in the development of AD [63], and some studies have suggested that accumulation of them can trigger mast cells and elicit exocytosis and phagocytosis [64,65], which supports our finding that the patients in S2 exhibit a higher proportion of mast cell activation. It should be noted that our results were based on the analysis of blood samples. This finding indicates that the impact of AD on mast cells can be reflected in the whole blood.

In our study, we utilized multiple machine learning approaches to establish and test the predictive model, respectively, with the aim of screening out the optimal model for AD diagnosis. Specifically, this strategy utilizes various feature selection algorithms (such as LASSO, random forest, etc.) to select features and evaluate the predictive capability of models via the AUC index. This strategy could well eliminate the bias that may exist in a single feature selection algorithm, which would improve the robustness and sensitiveness of the predictive model.

It is worth noting that this study has several limitations. Firstly, the absence of crucial clinical information such as survival time, survival status, cognitive test scores (e.g., MMSE, MoCA, CDR), and education level of AD patients, all of which are highly relevant to AD patients, restricts our ability to fully analyze the clinical features between the S1 and S2 groups. We expect to collect more clinical data on AD patients in our future work. Secondly, although MPPSS exhibits decent predictive performance no matter the testing data or independent validated data (including blood and brain datasets), there is still a lack of large-scale verification via prospective studies with large sample sizes. The MPPSS might be a valuable clinical tool aiding doctors in accurately diagnosing AD, especially in elderly patients, after rigorous evaluation and validation. Additionally, the lack of blood samples prevented us from conducting more stable external validation specifically for blood-based analysis. Nonetheless, we included samples from other brain tissues for validation, which further demonstrated the generalizability of our model. Finally, the functional role of the reliable MPP signatures we identified requires further molecular experiments. The employment of the SiMoA platform in measuring blood biomarkers associated with neurodegeneration [66], encompassing Aβ 42/Aβ 40, p-tau, NFL, and GFAP, has emerged as a promising approach for facilitating both the diagnosis and prognosis of AD. Rationally, there is a pressing need for additional research endeavors to explore the interrelationship between these neurodegeneration biomarkers and metabolic pathway markers, thereby allowing for enhanced comprehension of their biological implications in AD, which facilitates a better understanding of their biological significance implicated in AD.

Compared with existing neurodegeneration blood-based biomarkers, one potential advantage of the suggested metabolic pathway signatures over blood-based biomarkers for neurodegeneration is that they could be more practical for studying early-stage AD, as the metabolic biomarkers can be accurately measured in blood even in healthy individuals. Notably, the metabolic pathways-based approach also holds promise in aiding the diagnosis of other forms of dementia beyond AD [67,68].

In summary, we conducted a comparative analysis based on blood gene expression data between AD and non-AD groups. Characterization of the DEGs, and pathways associated with AD disclosed a potential correlation of metabolism with the onset and progression of AD. Based on blood transcriptome data, we constructed new metabolic markers, referred to as MPP signatures. Subsequently, we revealed the molecular subtype of AD based on NMF clustering and detected the differences within the AD subset distribution. Network analysis was applied to differential MPP signatures to detect the core metabolic network of AD. Eventually, we established MPPSS for AD diagnosis, which exhibited good performance on train, test, and validation datasets. Our study provides insights into the association between AD and metabolism, and MPPSS shows the important implications for AD diagnosis and treatment.

## Figures and Tables

**Figure 1 genes-14-01285-f001:**
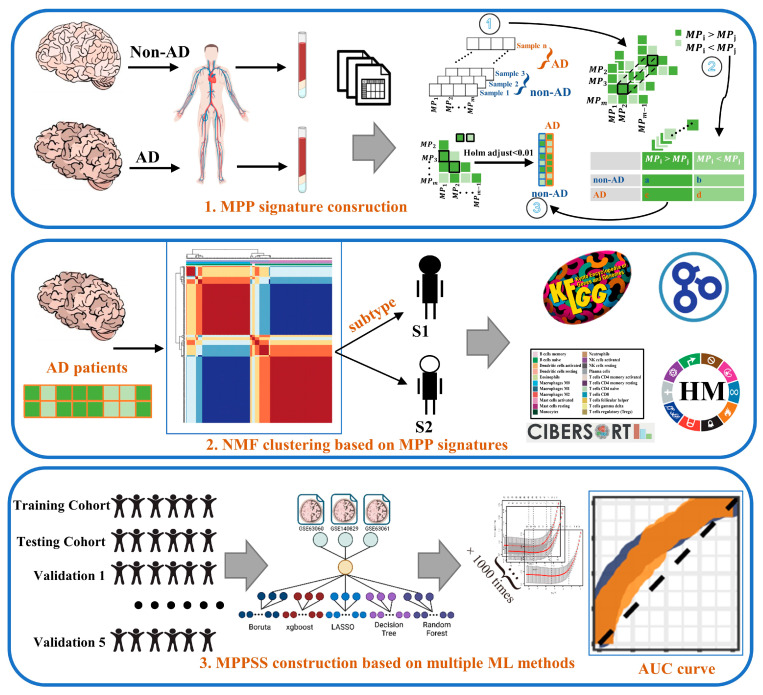
The flow chart summarizes the scheme performed to construct Metabolic Pathway Pairwise-based System scoring (MPPSS) for AD diagnosis. The icons of the human body, blood plasma, and brain images were utilized via BioRender (https://www.biorender.com/).

**Figure 2 genes-14-01285-f002:**
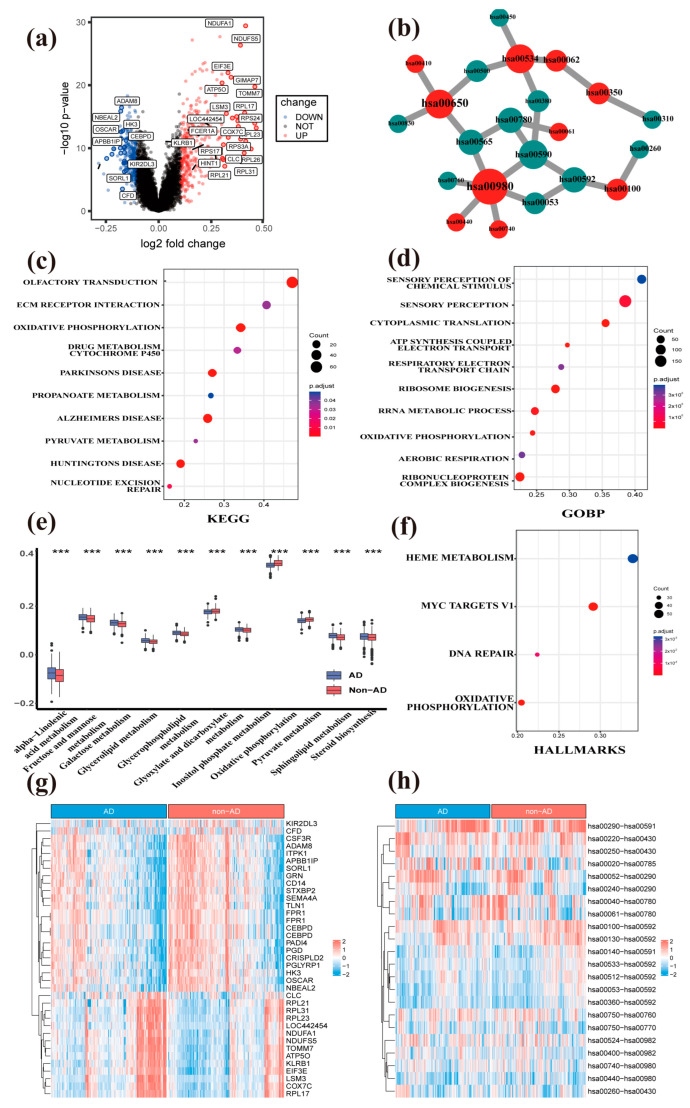
The metabolic hallmarks between AD and non-AD samples were detected by comparative transcriptomic analysis. (**a**) The volcano plot of DEGs between AD and non-AD groups. (**b**) The core metabolic network of significantly differential MPP signatures unveils important metabolic pathways. We selected the top 10 metabolic pathways ranked by the MCC algorithm and their neighbors for visualization. Hub nodes are also labeled by color, and the nodes with a deeper color have a higher rank. Green nodes are the neighbor nodes of hub nodes. (**c**) The functional annotation analysis of DEGs by the KEGG database. Over-Representation Analysis (ORA) has been used to calculate the enrichment level of KEGG from DEGs. (**d**) The functional annotation analysis of DEGs by the GO database. ORA has been used to calculate the enrichment level of GO from DEGs. (**e**) The boxplot of metabolic pathways between AD and non-AD samples reveals significantly differential metabolic pathways. (**f**) The functional annotation analysis of DEGs by hallmark pathways. ORA has been used to calculate the enrichment level of HALLMARKS from DEGs. (**g**) The heatmap of DEGs between AD and non-AD groups. We selected significant DEGs from all the available ones for visualization. (**h**) The heatmap of differential MPP signatures. We selected significantly differential MPP signatures from all available ones for visualization. hsa01040, Biosynthesis of unsaturated fatty acids; hsa00052, Galactose metabolism; hsa00780, Biotin metabolism; hsa00270, Cysteine and Methionine metabolism; hsa00760, Nicotinate and Nicotinamide metabolism; hsa00592, α-Linolenic acid metabolism; hsa00533, Glycosaminoglycan biosynthesis—keratan sulfate; hsa00590, Arachidonic acid metabolism; hsa00980, metabolism of xenobiotics by cytochrome P450; hsa00512, Mucin type O-glycan biosynthesis. The labeled asterisk indicated the statistical *p* values (*** *p* < 0.001).

**Figure 3 genes-14-01285-f003:**
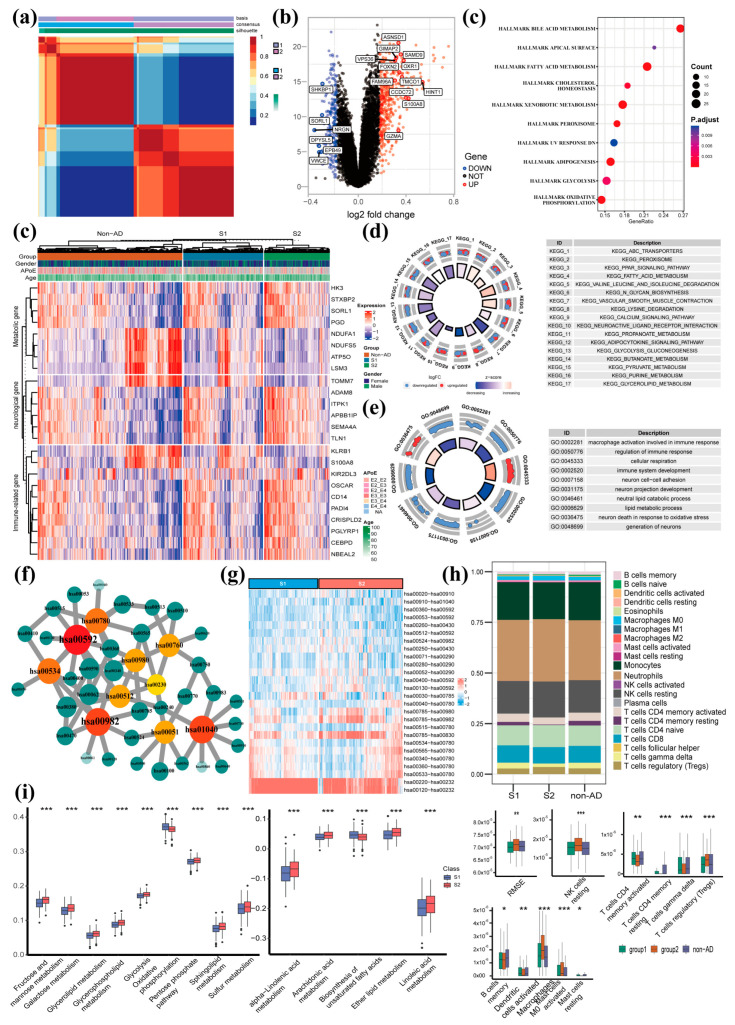
The characterization of DEGs, pathways, MPP signatures, and immune microenvironments between S1 and S2 groups. (**a**) The heatmap of NMF clustering based on AD patients. (**b**) The volcano plot of DEGs between S1 and S2 groups. (**c**) The functional annotation analysis of DEGs by hallmark pathways. ORA has been used to calculate the enrichment level of HALLMARKS from DEGs. (**d**) MPP signature-based clustering of the metaGEO dataset showed that DEGs significantly correlated with metabolism, nervous system, and immunity among S1, S2, and non-AD groups. (**e**) The functional annotation analysis of DEGs by the KEGG database. GSEA (Gene Set Enrichment Analysis) has been used to evaluate the enrichment of gene sets in KEGG. The red circle represents upregulation of the gene, while the blue circle represents downregulation of the gene. (**f**) The functional annotation analysis of DEGs by the GO database. GSEA has been used to evaluate the enrichment of gene sets in GO. The red circle represents upregulation of the gene, while the blue circle represents downregulation of the gene. (**g**) The core metabolic network of significantly differential MPP signatures. We selected the top 10 metabolic pathways ranked by the MCC algorithm and their neighbors. Hub nodes are also labeled by red, and the nodes with a deeper color have a higher rank. Green nodes indicate adjacency nodes between hub nodes. (**h**) The heatmap of differential MPP signatures, which we selected from all available ones between the S1 and S2 groups. (**i**) The immune infiltration analysis shows the profiles of infiltrating immune cell types between the S1 and S2 groups. (**j**) The boxplot of metabolic pathways between S1 and S2 groups. hsa00592, α-Linolenic acid metabolism; hsa00982, drug metabolism by cytochrome P450; hsa01040, biosynthesis of unsaturated fatty acids; hsa00534, glycosaminoglycan biosynthesis by heparan; hsa00780, biotin metabolism; hsa00980, metabolism of xenobiotics by cytochrome; hsa00512, mucin type O-glycan biosynthesis; hsa00051, fructose and mannose metabolism; hsa00760, nicotinate and nicotinamide metabolism; hsa00062, fatty acid elongation. The labeled asterisk indicated the statistical *p* values (* *p* < 0.05, ** *p* < 0.01, and *** *p* < 0.001).

**Figure 4 genes-14-01285-f004:**
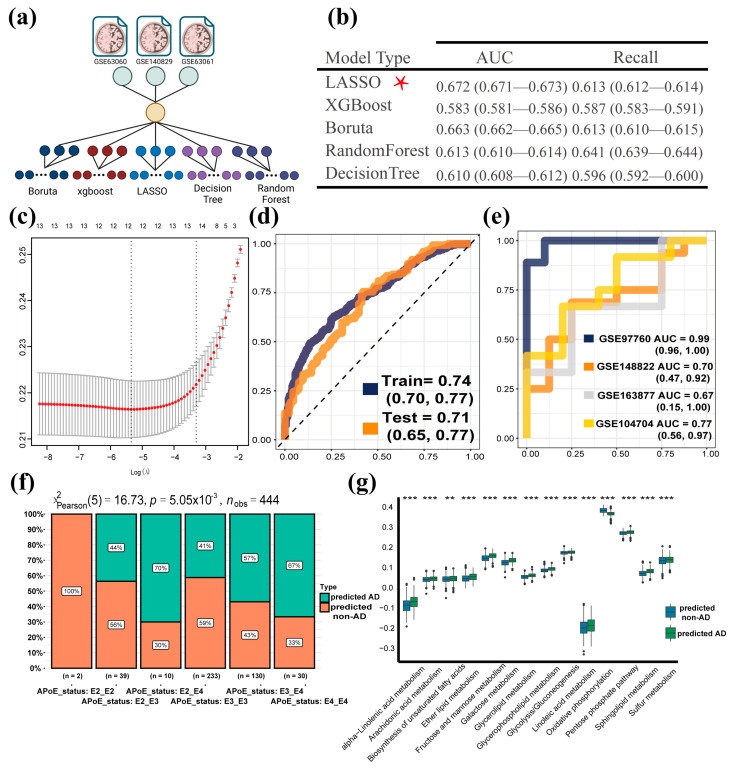
Establishment and validation of MPPSS for the diagnosis of AD patients. (**a**) The construction of MPP signature models based on multiple machine learning methods. The different color circles represent different machine methods. Dark blue represents Boruta, dark red represents xgboost, light blue represents LASSO, light purple represents decision tree, and dark purple represents random forest. (**b**) The performance among five machine learning methods on train and test datasets. The optimal method among five machine methods is marked by red asterisk. (**c**) The selection of MPP signatures by MPPSS; (**d**) The AUC curves of MPPSS on train and test datasets; (**e**) The AUC curves of MPPSS on four independent validation datasets; (**f**) The comparison of the APoE genotypes in two groups diagnosed by MPPSS on the metaGEO dataset; (**g**) The comparison of metabolic pathway activity in two groups diagnosed by MPPSS on the metaGEO dataset. The labeled asterisk indicated the statistical *p* values (** *p* < 0.01 and *** *p* < 0.001). ROC curves include the AUC value and the corresponding 95% confidence interval (CI).

**Table 1 genes-14-01285-t001:** The clinical information of 975 samples, including GSE140829, GSE63060, and GSE63061 datasets.

Characteristic	Group	AD Number (%)	Non-AD Number (%)	*p* Value
Samples	Disease state	488 (50.05%)	487 (49.95%)	-
Age	≤60	6 (1.23%)	5 (1.03%)	0.00336
	>60 & ≤70	133 (27.25%)	144 (29.57%)	-
	>70 & ≤80	222 (45.49%)	257 (52.77%)	-
	>80 & ≤90	126 (25.82%)	80 (16.43%)	-
	>90	1 (0.20%)	1 (0.21%)	-
Gender	Female	288 (59.02%)	282 (57.91%)	-
	Male	200 (40.98%)	205 (42.09%)	0.725
Race	Other Caucasian	31 (6.35%)	11 (2.26%)	-
	Western European	214 (43.85%)	171 (35.11%)	0.026405
	British	29 (5.94%)	48 (9.86%)	0.000267
	Irish	3 (0.61%)	2 (0.41%)	0.519052
	Any Other White Background	4 (0.82%)	4 (0.82%)	0.189355
	Any Other Asian Background	1 (0.20%)	2 (0.41%)	0.174688
	Unknown	206 (46.67%)	249 (46.67%)	-
APoE status	apoe_E2_E3	9 (1.84%)	30 (6.16%)	-
	apoe_E2_E2	0 (0%)	2 (0.41%)	0.982921
	apoe_E2_E4	7 (1.43%)	3 (0.61%)	0.009220
	apoe_E3_E3	78 (15.98%)	155 (31.83%)	0.201116
	apoe_E3_E4	78 (15.98%)	52 (10.68%)	0.000128
	apoe_E4_E4	26 (5.33%)	4 (0.82%)	0.00000294
	Unknown	290 (59.43%)	241 (49.49%)	-
Subgroups ^a^	S1	295 (54.92%)	-	-
	S2	193 (40.98%)	-	-

^a^ We used NMF clustering analysis to divide AD patients into two subgroups (S1 and S2) based on significant MPP signatures, which exhibit distinct activities of metabolism and immunity.

**Table 2 genes-14-01285-t002:** The contingency table of chi-square test to detect the correlation between AD and MPP signature.

Type	$MP_{i}{}^{\left(k\right)}\geq MP_{j}{}^{\left(k\right)}$	$MP_{i}{}^{\left(k\right)}<MP_{j}{}^{\left(k\right)}$
Non-AD	a	b
AD	c	d

**Table 3 genes-14-01285-t003:** MPP signatures used for LASSO model construction.

MPPS	Coef	Pathway Pairwise Function
hsa00100-hsa00190	1.0285978	Steroid hormone biosynthesis—oxidative phosphorylation
hsa00563-hsa00190	1.4211556	GPI-anchor biosynthesis—oxidative phosphorylation
hsa00534-hsa00190	1.0289191	Glycosaminoglycan biosynthesis-heparan sulfate/heparin—oxidative phosphorylation
hsa00900-hsa00190	1.1686399	Terpenoid backbone biosynthesis—oxidative phosphorylation
hsa00310-hsa00534	−0.6982631	Lysine degradation—glycosaminoglycan biosynthesis-chondroitin sulfate/dermatan sulfate
hsa00760-hsa00190	1.1373381	Nicotinate and nicotinamide metabolism—oxidative phosphorylation
hsa00531-hsa00860	0.1188049	Glycosaminoglycan degradation—porphyrin metabolism
hsa00513-hsa00620	1.3416181	Various types of N-glycan biosynthesis—pyruvate metabolism
hsa01040-hsa00190	1.0216412	Unsaturated fatty acid biosynthesis—oxidative phosphorylation
hsa00310-hsa00600	−0.8491503	Lysine degradation—sphingolipid metabolism
hsa00534-hsa00620	0.1976417	Glycosaminoglycan biosynthesis-heparan sulfate/heparin—pyruvate metabolism
hsa00310-hsa00531	−1.1770501	Lysine degradation—glycosaminoglycan degradation
hsa00051-hsa00860	0.4476219	Fructose and mannose metabolism—porphyrin metabolism

## Data Availability

Gene expression profiles were downloaded from the Gene Expression Omnibus (GEO) database (https://www.ncbi.nlm.nih.gov/geo/query/acc.cgi, accessed on 26 April 2023).

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
