# Peer review of "Metabolic Pathway Pairwise-Based Signature as a Potential Non-Invasive Diagnostic Marker in Alzheimer’s Disease Patients"

_genes, 2023, doi:10.3390/genes14061285_

Round 1

Reviewer 2 Report

This is a very thoroughly described paper about the application of metabolic pathway pairwise-based signature on AD diagnostics and machine learning as a tool for further analysis.

I have some minor concerns:

Table 1. It would benefit the paper to also include other diseases these patients have and hence, their possible influence on the results.

MPPSS 

Who was your control group? How did you check if output of it was correct?

2.6 Please elaborate more on AD subgroups.

Reviewer 3 Report

This paper presents a study to construct metabolic pathway pairwise signatures to characterize the interplay among metabolic pathways in Alzheimer’s disease patients. Authors applied various bioinformatic approaches/tools to investigate the molecular mechanism behind the Alzheimer’s disease. Authors shared interesting findings, while there are some questions as outlined below.

1-     Authors applied he early integration across three cohorts (SVA and ComBat), while not shared the details of integration. Please add more details (including number of estimated batches, visualization figures, PCA, etc.) before and after removing the potential batch(es).

2-     It is important to select train (i.e., metGEO here) and validation cohorts. Are there any specific inclusion (or exclusion) criteria to select train and test cohorts? Please clarify this part and why 3 train and 5 test?

3-     Tale 1 needs to be improved:

a.      How the cut-offs for the age were defined?

b.     S1 and S2 sub-types need to be defined.

c.      Is it possible to generate the table based on outcome (AD and non-AD) for each clinical variable? Then, assess the association of the variables with outcome of interest (AD vs non-AD)?

4-     Choosing the threshold 70% and 30% for train and test in ML approaches is challenging. How were these cut-offs selected and why not considering k-fold CV?

a.      What is the common performance metric across ML approaches, Fig 4b? Please add more details.

b.     I assume 1000 times iteration at training steps, then the reported AUC is mean AUC? If yes, please add 95% CI or SD across figures and manuscript. I couldn’t follow.

5-     What is the specific method to do the enrichment analysis? Based on shared figures, I assume it is over-representation. Please clarify and add more details.

6-     I’m wondering to know why authors did correction for multiple tests by controlling FWER (e.g., Holm method) instead of FDR (e.g., BH or FDR)?

7-     All the figures’ resolution needs to be improved and please add more details for the captions to be more informative. As an example,

a.      The box plots only have *, **, or *** while they did not define before.

b.     Fig 2a, x-axis is logFC?

8-     To have reproducible research, it is important to share the codes specially for ML approaches. Please share the GitHub link(s).  

Round 2

Reviewer 3 Report

Authors addressed all the comments. I have only two minor comments as follows

1- For Table 1, please add the appropriate statistical test in method section

2- The GitHub link I couldn't get access. Please make sure the shared link in manuscript is public and available for readers.

3- All the reported AUC values across the manuscript (e.g., in abstract) better to be reported as AUC (95% CI). 

I'd like to ask Editor to consider these two comments and make decision. 

Thank you! 
